# Belonging to the City: Alliances between Community Art and Diaconia as a Means to Overcome Segregation in a Gentrifying Neighbourhood in Amsterdam

Erica Meijers

Protestant Theological University, 1081 HV Amsterdam, The Netherlands; p.e.m.meijers@pthu.nl

**Abstract:** Between 2019 and 2021, volunteers of a local Protestant congregation in Amsterdam, professional artists, and (other) local residents organised the interactive exhibit *A(t) home in the Staats*. In this project, community art and diaconia joined forces using multidisciplinary methods to strengthen relations in the neighbourhood and to discern issues of belonging and lines of division in the changing neighbourhood. The project was situated at the intersection of an "up and coming" neighbourhood and a shrinking congregation. By analysing the exhibit, this article contributes to the development of creative, arts-based research methods in diaconal studies. Within this approach, art is never a mere illustration or a vehicle for reflection but rather a generator of knowledge. The central question is: how can alliances between community art and diaconia contribute to overcoming segregation in urban contexts? This question is informed by the process of gentrification and the search by city churches for ways to engage with urban changes. After the introduction and methodological reflections, the article describes the background and practice of the project, followed by the outcomes of the interactive exhibit. It concludes by answering the central question and mapping theoretical and practical challenges concerning alliances between art and diaconia in urban contexts.

**Keywords:** diaconia; community-art; gentrification; urban theology; urban mission

## 1. Introduction

This contribution explores possible alliances between art and diaconia in order to strengthen local communities in cities. In such places, capitalisation and gentrification increase segregation between successful local residents on the one hand and those who are not able to profit from the economic developments on the other.

Art is akin to religion in the sense that both express lived realities and imagine other worlds (Adams 2012). This article deals with artforms that are more specifically related to diaconal practices. Diaconia, performed by either ordained deacons, volunteers, or diaconal professionals, is the Christian practice of engaging with people in precarious positions in our societies, with the intention to "create a space for new breathing, for liberation, for dignity and for comfort" (Ampony 2021, p. xxix). As such, diaconia is a central dimension of the Christian faith and a mark of the church (Dietrich 2019, p. 1).

The work of artists engaging with precarious groups usually moves outside the formal artworld of galleries, expositions of famous artists, and auctions. It is more likely to be found on the streets than in museums. Even though their work has intrinsic value, these artists do not believe in *l'art pour l'art*, but rather want to contribute to social change. They not only aim at imagining worlds but also want to create new realities. Therefore, this type of art is seldom an expression of individuals; instead, it is usually a communal effort (Brueguera 2010).

There are many different names for this kind of art: engaged art, political art, community art, collaborative, and participatory art (Malzacher and Staal 2021).[1] They are all

related to what has been called the "social turn in art" (Bishop 2005). In this article, I follow community artist Francois Matarasso (2019, p. 48), who uses the term *participatory art* to indicate the whole river of collaborative practice in which artists work with others to make art and *community art* to indicate an approach characterised by the aspiration of emancipatory social engagement. This description of community art is in line with the aim of diaconia to contribute to liberation and dignity, as described above. According to An van den Bergh (2012), community art helps to imagine other worlds, empower social relations, connect different people, and renew society. Matarasso (1979, p. 83) states that cultural participation strengthens both individuals and local communities by developing self-confidence, creativity, personal growth, and social cohesion by helping especially minorities and excluded groups to develop their own stories. Albert Dzur (2010) underlines the importance of a democratic space in which citizens can learn to handle changing societies through the cooperation of professionals and laypeople. Sharog Heshmat Manesh (2022, pp. 242–48) develops the view of the cultural professional as "a witness", based on collective reflection by professionals and lay people on their roles in public space. "Witnessing", a concept he borrows from philosopher Kelly Oliver (2001), builds on the strength of storytelling in a community and contributes to the restoration of injustice and pain by making it visible. All these theories take as their starting point the commitment to strengthen social development and human flourishing. In diaconia, just like in community art, connecting people is a core business (Meijers and Tromp 2022), and it is often directed towards the creation of "just and inclusive communities".[2]

The value of community art for diaconal practices has received little attention in diaconal studies so far.[3] This article contributes to the development of creative, arts-based research methods in diaconal studies, as introduced in practical theology by Clare Luise Radford (2020) and others (Boursier 2021; Walton 2019), and reflects on the role of art in diaconal practices. My leading question is: how can alliances between community art and diaconia contribute to overcoming segregation in urban contexts?

In order to investigate alliances between diaconia and community art, we will have to turn to specific situations and lived realities since, as one of the great artists of the twentieth century, Bertold Brecht, had written in big letters over his desk in his Danish exile, "truth is always concrete" (Malzacher 2014, p. 5). I therefore dedicate this article to a project in a gentrifying neighbourhood in Amsterdam, in which community art and diaconia worked together in the exhibit *(T)huis in the Staats*. In Dutch, the title is a wordplay: "thuis" meaning "at home", but "huis" meaning a house and referring to the church building. For the participating Protestant congregation, this implied a question: how could the church belong to the neighbourhood and yet offer a place to people in the neighbourhood? In English, this can be translated as *A(t) Home in the Staats*.[4] *Staats* refers to the name of the neighbourhood: Staatsliedenbuurt, where the names of the streets are called after statesmen. During the project (2019–2021), artistic and creative methods were deployed to generate knowledge regarding the relation of the participants to their neighbourhood and to express and define this relation anew to overcome (increasing) segregation.

Diaconia in this context is understood as community development. Just like "community art", "community development" cannot be easily defined. It takes on a different form depending on the context in which it is used. Tony Addy (2022, p. 15) describes five different types: participatory and inductive, participatory and critical, organising as an approach, institution-based, and market-focused. *A(t) home in the Staats* has features of the first three types: it was non-directive, promoting participation and self-determination; it started from the strengths of the community and implied asset-based strategies. Stories and dialogues played a central role, and the project was founded on the awareness that socio-economic and political realities need to be transformed. The fundamental equality of all people and their empowerment were core ideas (Addy 2022, pp. 19–25). It brought people together to build relationships across diversities and decrease segregation in the neighbourhood.

This contribution consists of three sections, preceded by some methodological reflections concerning arts-based methods, collaborative action research, and my own position in the project. The first section will describe (1) the urban and ecclesial backgrounds of the project, (2) The aims, participants, and approach, and (3) the artistic methods and activities during the exhibit. In the second section I will present the outcome of the exhibit. The third section will map theoretical and practical challenges concerning the collaboration between art and diaconia in urban contexts.

*Methods and Positionality*

Claire Luise Radford presents arts-based methods as a way of generating knowledge through the practice of the creative arts, "rather than using artistic works as illustrations of theological points or a starting place for spiritual reflection" (Radford 2020, p. 61). This type of research resists instrumentalising art. I therefore speak of alliances between art and diaconia instead of arts-based methods for diaconia. Moreover, this seems more appropriate in a situation in which diaconal practices depend on collaboration with others in order to be effective. This is the case in the highly secularised city of Amsterdam, where only fifteen percent of the population identify as Christians.[5] The fact that the project *A(t) home in the Staats* was small does not hinder the exploration of collaborations between diaconal and artistic methods. The project reached approximately 500 people in a neighbourhood of 12,000 inhabitants, of whom around 100 participated more than once or were actively involved.[6] It was imbedded in ongoing practices, and can be regarded as just one episode in a durable commitment to overcoming segregation in the neighbourhood.

The use of arts-based methods is inscribed in the tradition of theologies that are embodied, material, and emerging from everyday life and that take us "further into what is complex, contradictory, and uncertain in our attempts to trace the sacred in our practices of liturgy and learning, protest and peace-making, and everyday life" (Radford 2020, p. 62). The uncertainty and complexity of social processes are well known in both diaconia and community art.

The project can be further described as collaborative (artistic) action research, defined by Henk De Roest (2020, p. 98) as a type of research that "focuses on needs and on the identification and communal definition of concerns. Action research creates an opportunity to address the real problem of a 'community of practice'". De Roest underlines the impact of this type of research, while Donald Schön suggests that solving problems is not the most important aspect of this research. According to Schön, the real value lies in formulating the problems, selecting characteristics of a specific context, (re)directing attention towards these, and creating a coherent framework within which an assessment of possible solutions can be made (Schön 2006, cited by Heshmat Manesh 2022, p. 83). This is what *A(t) home in the Staats* aimed to do. De Roest (2020, pp. 208–11) differentiates between participatory and collaborative research. Firstly, the study seems to be more firmly in the hands of professional researchers, who are given an empowering or educative role. However, in collaborative research, no one is an "object" of study; instead, all are both active subjects and participants. In *(A)t home in the Staats*, there were differences between participants (I will return to this point later), but there were no external researchers. I was no outsider myself. I am a long-term, active member of the local congregation and have lived in the area for more than 35 years. As a member of the initiating and organising team, I took notes at all meetings and documented the whole process. I coordinated and, to some extent, trained the interviewing group, interviewed myself, moderated the opening of the exhibit, and took part in the closing political debate as a representative of the church. I was part of many behind-the-scenes deliberations on social relations, organisational problems, and the interpretations of the project. A year after the end of the project, I organised a focus group discussion with the initiating group, during which an earlier version of this text served as a basis for looking backwards and forwards, and for discussing the features of our process together.[7] Those who have commented on this article are mentioned with their full names, while all other participants will be anonymised since they only gave consent to distribute

their interviews locally. This article is part of a collective process of action and reflection in the sense that it tries to illuminate the experiences and methods of the project in order to act anew. However, I am responsible for the (creative) process of interpreting, selecting the material, and composing this representation of our common experiences. In this article, I focus on alliances between art and diaconia, but I will offer another perspective elsewhere.[8] Regardless, my insights clearly cannot grasp the full experience of the exhibit. At their best, they represent just another step in the ongoing process of learning how to be Church in the neighbourhood.

## 2. Backgrounds and Framework of *A(t) Home in the Staats*

*A(t) home in the Staats* situated itself at the intersection of an "up and coming" neighbourhood and a shrinking congregation. To provide more insight into the background of the research question, I will sketch a brief historical overview of the neighbourhood and the engagement of the local congregation. After that, I will introduce the aims, participants, and approach, and thirdly, the artistic methods and activities of the project.

### 2.1. A Shrinking Congregation in an Up-and-Coming Neighbourhood

The *Staatsliedenbuurt* was built at the end of the 19th century. During its first decennia, a mix of working-class people and civil servants such as policemen and firemen lived in the neighbourhood. Residents who could would leave for better areas of the city. Slowly, the quarter became known for its poor and rebellious inhabitants. Socialists, communists, and anarchists organised manifestations, sometimes fighting in the streets. Unemployment was high, and people struggled to make ends meet. In the 1920s, the Nassau church was built based on a neo-Calvinistic congregational model. During the 1970s and 1980s, the neighbourhood became known for its accumulation of social problems. The houses were in bad shape; they typically contained three floors with apartments of about 40 square metres. Many of them were abandoned, while housing shortages in Amsterdam were high. In these circumstances, social protests against speculation occurred, and the *Staatsliedenbuurt* became one of the hotspots of the squatters' movement. They improved the houses and founded many organisations, such as the tool lending service *De blauwe Duim* (The Blue Thumb), which still exists. Nevertheless, the neighbourhood also suffered from drug addicts and criminality related to the declining squatter movement.[9] In those years, Amsterdam was in a deplorable state in the eyes of economists and sociologists. Due to the end of the industrial era and globalisation, unemployment was high, and many people left the city. In the eyes of some people, the *Staatsliedenbuurt* became a no-go area (Milikowski 2018, pp. 28–29). It was in this period that the Nassau church, which had just fused with another reformed church in the neighbourhood that had been torn down due to secularisation, chose to open up to the neighbourhood. The congregation committed to the ideas of urban mission: a church could not be Church when it isolated itself from the increasingly non-Christian community it was part of.[10] One of the ministers formulated the engagement as follows: "God's Spirit and mercy are present in the neighbourhood and go ahead of us. God became human, not in an abstract human, but in one who served those who were hurt and who, as a consequence, was executed like a slave. As a church, we try to follow God's special attention for the broken and vulnerable people in our relationship with the neighbourhood" (Irik 1995, p. 116.).[11]

Many activities grew out of this commitment, starting with solidarity with the squatters and other organisations. No one was excluded from cooperating. A small restaurant under the Persian name "Filah" (wellbeing) was opened by a member of the communist party who later became an active member of the congregation. This became popular among homeless people and others living on a low income. A volunteer took the initiative of opening a second-hand shop, and the church opened a walk-in centre for all those needing a moment of peace or looking for counselling and support. For funding reasons, these activities were brought together in the foundation *Kerk & Buurt* (Church & Neighbourhood), which has since added many more initiatives. Today, the foundation has many volunteers

with no ties to the church or the Christian tradition. Its goal is to enhance the quality of life and social cohesion in the neighbourhood with a keen eye for the uniqueness of all participants and volunteers.[12]

Change came to the city during the 1990s: the upcoming creative economy (Florida 2002) and information age (Castells 1996, 1997, 1998) needed a network society where knowledge could be shared and multiplied. The messy artistic and social spaces of Amsterdam provided these networks. The city started to grow again and attracted private investors, young urban professionals, and project developers. Desolated areas, transformed by artists into trendy and creative places, were developed by project developers with private and sometimes public capital; prices went up, and most artists were unable to stay. This model, which initially started spontaneously, was then intentionally applied by the city council (Stuart and Grote 2017). The government attracted entrepreneurs and facilitated the sale of social housing in the historical centre and especially the neighbourhoods surrounding it, the so-called nineteenth-century ring (Uitermark et al. 2023), where the *Staatsliedenbuurt* is situated. A new creative and wealthy class helped Amsterdam boom again. The increasing number of tourists in the city contributed to this development. The downside of what has been called "the resurrection of Amsterdam" (Milikowski 2018, p. 11) has been analysed by social scientists: Tourism causes the "Disneyfication" of the city[13], public space shrinks due to privatisation, and the increasing presence of foreign capital drives up prices, leading to spatial inequality (Uitermark et al. 2023). One of the major effects of this development is a process called gentrification. Urban geographer Cody Hochstenbach (2017) describes it as follows: first, the less fortunate are pushed out of the city centre and its surroundings, then the middle class starts to struggle as well. This leads to families leaving the city, schools closing down, the downgrading of social networks of parents and elderly people, the arrival of web shops, ever-faster grocery deliveries for the higher middle class, which undermine small neighbourhood shops important for the precariat, the declining quality of public space, etc. The Dutch-American sociologist Saskia Sassen (2014) has analysed processes of gentrification and capitalisation all over the world and observed a reality of expulsion at the heart of them: the dynamics of money and real estate exclude more and more people. Macroeconomic figures suggest wealth, but the real costs are made invisible: "The idea of globalisation suggests that everybody is connected, and individuals have access to a bigger zone than ever: physically and digitally. However, in reality, there is less space: less for food, less clean water, less public space, etc. ( . . . ) In the end, people become commodities. Those who do not generate money are useless and overlooked". "Artists", she thinks, "because of their special creative abilities, can make the expulsed visible again" (Meijers 2015).

The *Staatsliedenbuurt* is also affected by these developments. Social housing shrank from 61 percent in 2012 to 47 percent in 2021.[14] People with low and lower middle-class incomes have access to these houses. Today, 24 percent of the houses in the quarter belong to the "free rent" sector, with much higher prices, and 29 percent are owner-occupied homes. The value (and price) of these houses has gone up by 89 percent between 2013 and 2022.[15] Since the waiting list for the social sector is around 13 years, new residents, and children of old residents, pay a lot more to acquire homes in the neighbourhood. As a consequence of rising real estate prices, small shops struggle to survive and are often replaced by chains. The cultural development of the local park, the *Westerpark*, attracts many tourists who were never seen in this area before. Their presence has changed the types of restaurants and other facilities in the neighbourhood, and festivals in the park have become a nuisance to residents who do not use these facilities.[16] The mix of "successful" newcomers and old inhabitants who do not profit from the new economy causes increasing segregation in the neighbourhood. This is well illustrated with an (rather positive) anecdote: The local police received a complaint from a newcomer in the neighbourhood who had just bought an apartment for 600,000 EUR and was annoyed by the presence of a "shabby-looking" man selling the street paper in front of his house. The policeman answered: "Oh, that is Ali, he is part of this neighbourhood just like you. You better get used to him".[17]

The congregation of the Nassau church is trying to navigate this situation. It is losing people, partly because some move elsewhere to better or cheaper areas, but also because the congregation is getting older. There are very few new church members; most children leave the congregation when they grow up. There are still many people connected to Church & Neighbourhood, but the connection to the church has weakened, and volunteers and participants of the foundation often have fewer active relations with the congregation. The church no longer has enough official members to afford a full-time or even half-time minister and is struggling to find theological professionals to support the congregation for even a few days a week. The future of the congregation and its mission of contributing to wellbeing in the neighbourhood also depend on whether or not the (expensive) monumental church building can be used as an actual home in the neighbourhood.

In the gentrifying neighbourhood, the congregation started to reformulate its mission in 2016. In the 1980s, the focus was on commitment to the suffering in the area. Now, the importance of connecting people of different backgrounds is underlined by the church council: "people from different social classes meet at eye level as images of God; biblical stories are read from the perspective of this community, in which there is no difference between those who are members of the church and those who are not", since "God can be known through the other".[18]

New initiatives tried to form connections with newcomers in the neighbourhood. *Sjiek and Sjofel* (Classy and Shabby) organised vespers around exhibits of individual artists and held Christmas cafés for children.[19] This was a difficult process, and the minister initiating these projects had to leave the congregation for financial reasons. However, the congregation decided to continue the efforts: "We want to observe attentively ( . . . ) looking for unexpected connections which increase humanity and solidarity in the city. We want to investigate which desires live among the people in the neighbourhood and among ourselves".[20]

In this spirit, a small group started to organise "exposure sessions": being present in the streets with attentive eyes and ears to make new connections and discover what is important to the residents of the area.[21] The "exposure group" dedicated itself to developing and strengthening an attitude of receptivity to the neighbourhood as a form of Christian presence.[22] In the 1980s, this was conducted by professionals and that is still usually the case in other contexts. Yet, here it was carried out by members of the congregation, with some support from a professional of the *Kor Schippers Institute*, who specialised in "experienced based learning" as a form of urban mission.[23] For several Saturdays, small groups went into the neighbourhood without an agenda and with no specific task but to be there for a few hours. They walked around, sat in the park, and talked to strangers or old acquaintances. Afterwards, they shared their stories and their experiences. After a few rounds, a new step was taken: key people in the neighbourhood were interviewed: a creative entrepreneur, a school director, a community artist, and a few others.[24] From these experiences and relationships, new ideas arose. One of them led to the project *(T)huis in the Staats*.

## 2.2. Aims, Participants and Approach

In "emancipatory action research", Henk De Roest (2020, p. 196) writes that communities of practitioners take joint responsibility for the development of practices based on mutually shared and articulated values. Some of the questions asked included: What do we want to achieve? What are our habits and customs, and can we revise them? What are the control structures that prevent us from being change agents in our own professional practices? These were exactly the types of questions discussed in the fall of 2019 by a small group gathering around my kitchen table over a bowl of vegetable soup. They were artists, cultural professionals, members of the local congregation, and social activists (or a mix of all these).[25] We brainstormed to plan an art project that would gather stories from the neighbourhood to find out what was going on in the area while at the same time stimulating encounters between different generations, between newcomers and long-time residents, and between churchgoers and other inhabitants to overcome the deepening gaps caused by

gentrification. One of the participating artists showed us a picture of a container of building waste. Its symbolic power helped us formulate our project. The container symbolised the ongoing processes of renovation in the neighbourhood and, thus, the effects of gentrification. It also showed the waste itself: what and who disappear in this process of upgrading? Did all inhabitants (and those without residence who regarded the neighbourhood as one of their important whereabouts) still feel they belonged to the neighbourhood? Urbanists Jan Willem Duyvendak and Fenneke Wekker have shown the importance of feeling at home for improving the quality of life in neighbourhoods: "Feeling at home and experiencing interconnectedness are seen as preconditions for a "good" and liveable neighbourhood and city" (Duyvendak and Wekker 2016, p. 23). Within the politicised debate on home and identity (Duyvendak 2011), we did not search for a clear-cut neighbourhood identity or defend our home against any change, but we wanted to articulate the unseen and unheard and find out what was simmering below the surface. Who profited from the actual changes and who did not? How did this affect the neighbourhood community? This required an open process in line with the attitude of exposure: to encounter strangeness in our own well-known streets.[26]

Starting with a period of historical research and interviews with neighbours, an exhibit was organised during which the gathering of stories could continue through workshops and meetings. The issues were then discussed with local politicians, activists, and others in a closing debate. The project was prepared and carried out by individual artists and cultural professionals with an attachment to the neighbourhood and by two groups: the exposure commission of the Nassau church (with the support of the church council) and the organisation Church & Neighbourhood.

The group consisted of around twelve people in a changing composition throughout the project. It had several characterising features that determined its outcomes.

The first characterising feature was the diversity of the organising group. Not only were there two organisations involved, but the participants were of various backgrounds regarding their worldviews and religions, gender, and sexual orientation. Most of them were well educated and some had professional backgrounds in community development. They were all white, but in the wider circle, people of colour participated as well. Motives were also diverse since church members had theological reasons for their desire to renew connections between the church and the neighbourhood, while others were more interested in the social tissue of the neighbourhood itself, political change, or artistic methods and products. A participating student studied the project as an assignment in oral history. However, all initiators shared an interest in exploring issues of belonging in a gentrifying neighbourhood. Apart from the organising team, other participants in the project were: professional and non-professional artists, writers, politicians, homeless people, students, pensioners, activists, schoolchildren, former squatters, old and new inhabitants, privileged persons, people in precarious situations, and volunteers from both the neighbourhood and the Protestant congregation.

The second characterising feature was the non-hierarchical cooperation between the professionals and volunteers, or as Tony Addy (2022, p. 20) says, "the organisation was non-directive". There was little money involved, and all money was raised outside the participants' own organisations, which rather contributed materials, time, their building space, and employees/volunteers. We decided to manage the project together and divided ourselves into smaller groups responsible for various parts of the exhibit: an interview group, a history group, a planning group, a funding group, and a public relations group.[27] We gathered regularly to exchange information, but most decisions were made by the subgroups themselves. There were moments of joy and irritation; we had to learn to give space to each other's particular proprieties and capacities, and to accept some degree of chaos, which sometimes caused frustration. Informal power relations within the group sometimes led to misunderstandings or stress.

The third characterising feature can be defined by the motto: process over product. However, some leaned more towards the product and others more towards the process. The

core of the project was the sharing of stories, the meetings between neighbours outside their own bubbles, and the strengthening of relations within the group. The actual exhibit (lasting seven weeks) was "only" a moment in that ongoing commitment to the neighbourhood and all its inhabitants.[28]

The fourth characterising feature was the stamina of all participants. Even when there were conflicts and the organisation seemed chaotic and stressful, the group carried on. This was due to familiarity with this way of working, strong, long-lasting personal and collegial relations between members of the group, and the support of the individual networks and communities involved. Extra (temporary) volunteers could be mobilised if needed.

After a few months of preparation, the COVID-19 crisis interrupted our workflow. Two of the artists involved also had to leave due to personal reasons. Regardless, we decided to carry on with the possibilities we had. The COVID-19 crisis forced the group deal with more interruptions, messy circumstances, and difficulties raising funding. Creativity, improvisation, and flexibility, which had always been part of working in a changing neighbourhood with little money and vulnerable people, became even more vital. COVID-19 underlined the need to manage chaos and keep on working with whatever and whoever was available. We had to learn to trust the process instead of controlling it. Immediately after the exhibit, many participants said they had experienced difficulties with the "poor organisation" or the lack of coordination: "It is a nice motley crew, very different people. That stimulates the creativity. We have grown during the project and got to know and appreciate each other. We missed a coordinator, somebody who directed traffic".[29]

Yet, a year later, many of the initiating participants had changed their views: this open, and therefore sometimes chaotic, way of working had been, and still is, necessary for reaching our goals, as long as everybody is valued and there is continuous reflection on what is happening. Since there was nobody to complain to, everybody took responsibility; there were open spaces in which people took their own initiatives and mobilised their own networks. The shared values, long-time connections, and receptive attitude towards the neighbourhood created a feeling of (artistic) freedom and ownership, a collective direction ingrained within the interaction itself. "If we would have been product-oriented, we would have had a fancy exhibit, but the meandering process gave a much richer harvest: the connections made might be fewer, but they are sustainable".[30]

This way of working is not only typical for diaconal, participatory, and critical community development (Addy) but, according to Francois Matarasso, it is also typical for community art, which he defines as "restless art": " . . . it is an ever changing, moving form of art, with changing participants, goals and always talking back to what the city is offering. It is full of tensions and ambiguities, arising from co-creation by professional artists and others, crossing borders between art, politics, social activism and other while searching for "democratic sense-making" as a hopeful way of making life better. It is an art which is never content or finished, in short: restless" (Matarasso 2019, p. 17).

Finally, the exhibit took place between lockdowns in October and November 2021.

### 2.3. Artistic Methods and Activities

Fundamental to *A(t) home in the Staats* was storytelling, which continued during all three phases of the exhibit: the preparation, the exhibit itself, and the closing debate. Artistic methods were used to stimulate the sharing of memories, experiences, and expectations concerning ways of belonging to the neighbourhood. In the preparatory phase, twelve local residents were interviewed. They were selected to reflect the diversity in the neighbourhood. Criteria were: gender, age, income, lifestyle (living as a family, single, in a network, isolated, or on the street), and how long they had been living in the neighbourhood.[31] Apart from being interviewed, the interviewees were photographed at a favourite spot in the neighbourhood by Peter Valckx. Their portraits were displayed in the church during the exhibit and their stories and photos were published in a one-time magazine, which functioned as the catalogue of the exhibit (Dorssers and Meijers 2021). During this first phase, some participants dived into the archives of the city, the church, and the

neighbourhood, and collected texts and images which constituted the raw material for a series of five collages made by one of the participating artists, Philippe Velez McIntyre, who was assisted by a few others from the group. The first collages give an impression of the period between 1880 and 1930 (Figures 1–3), the second tells a story of the 1950s (Figure 4) and the third shows the 1970s and 1980s, the time of the squatters (Figures 5 and 6). The next collage portrays the work of the church, of Church & Neighbourhood, and many of the people involved (Figures 7 and 8). The last collage focused on the housing issues due to gentrification and used a recent demonstration against the lack of affordable houses in the city as a theme (Figures 9 and 10). An extra collage shows the beginnings of the church, which was built after a design competition for architects (Figure 11).

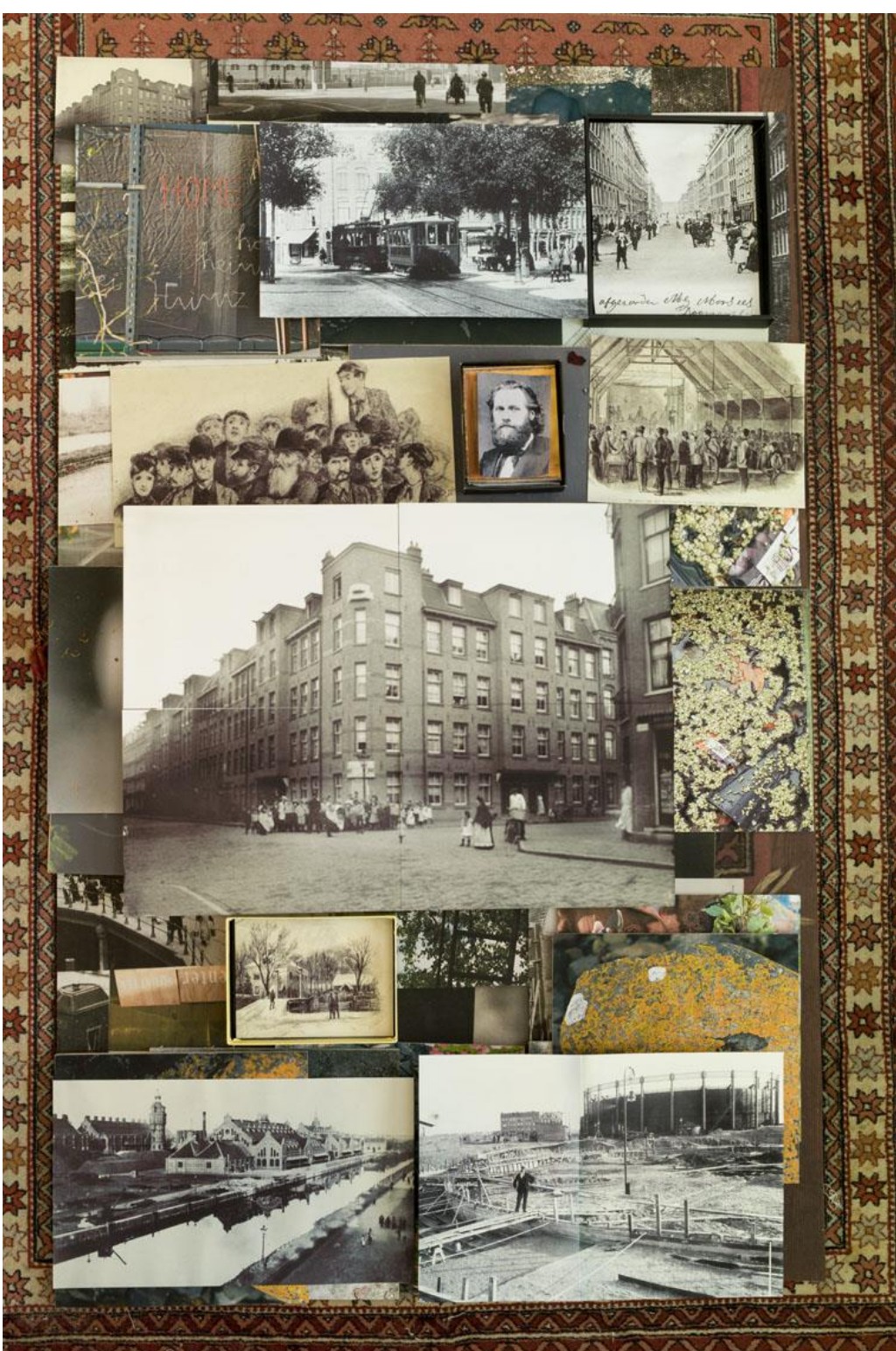

**Figure 1.** Collage on the beginnings of the neighbourhood around 1900. In the centre, a picture of rev. Domela Nieuwenhuis, who left the church because of its lack of solidarity with the working class to become an anarchist politician. His great-grandson was well known in the neighbourhood for organising crazy cultural and political festivals.

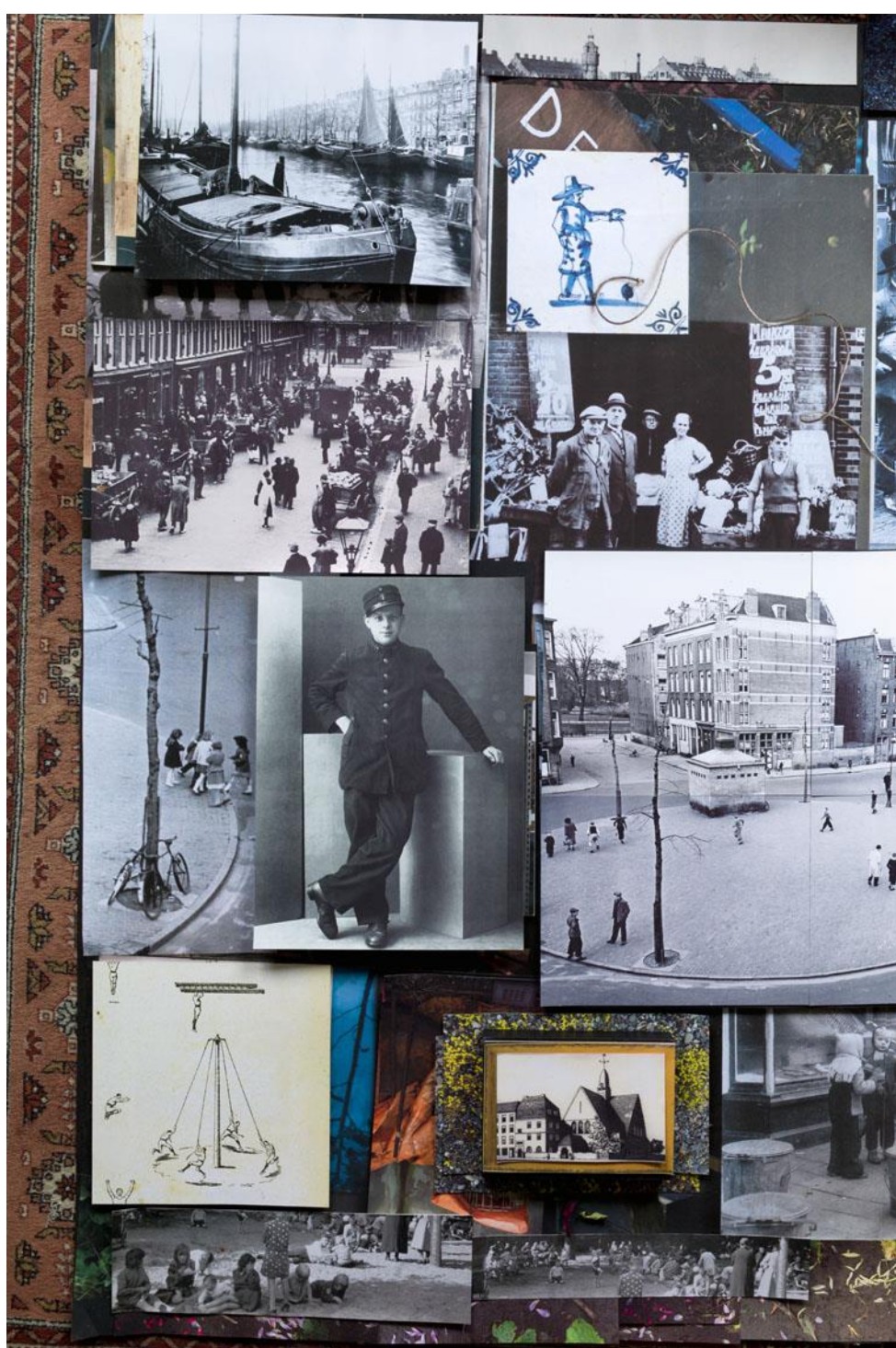

**Figure 2.** First half of the collage with photos from the 1930s. The neighbourhood was inhabited by anarchists, communists and so-called brass button civil servants, which showed they were more than just poor working class.

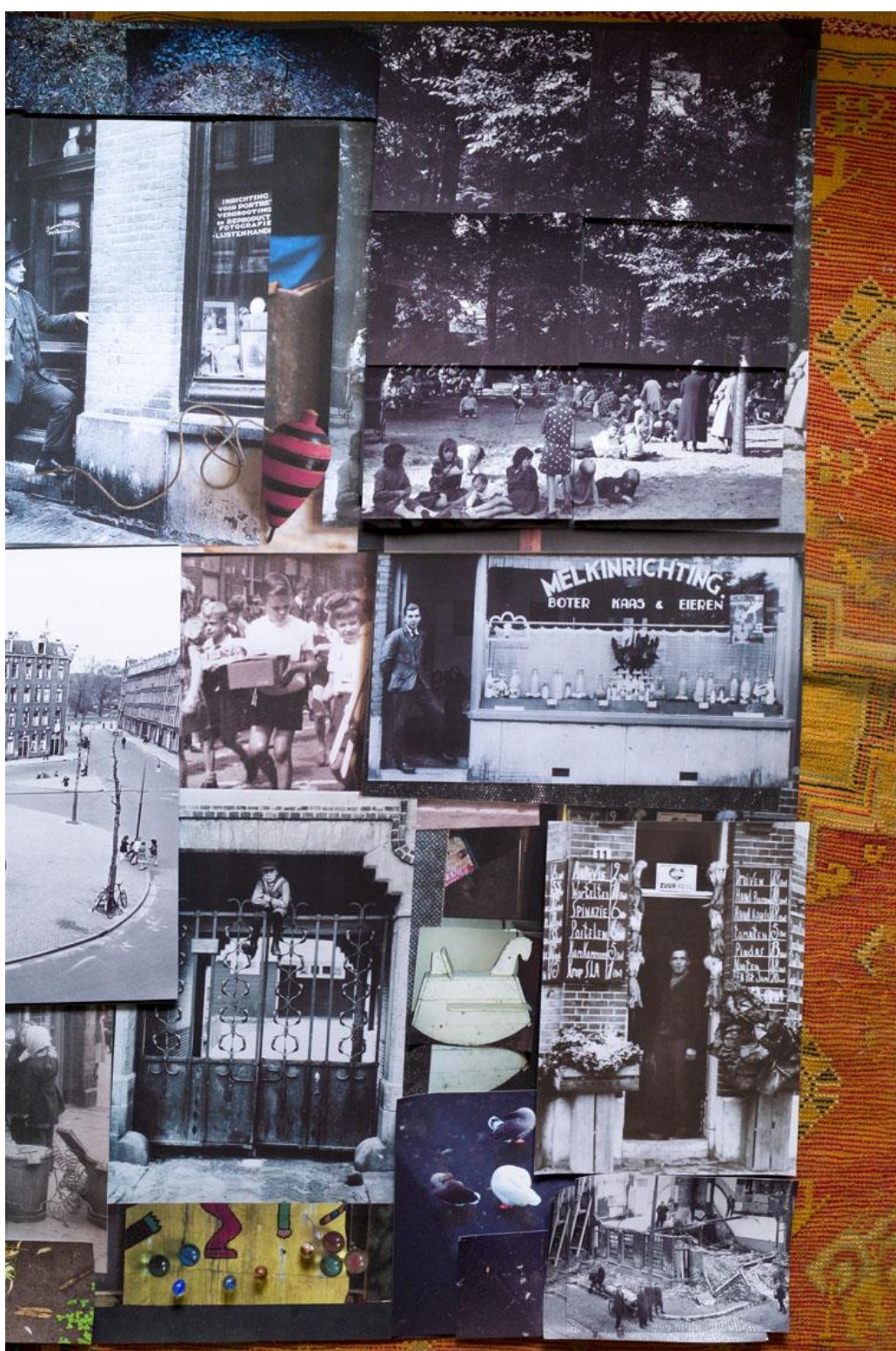

**Figure 3.** Second half of the collage of the 1930s.

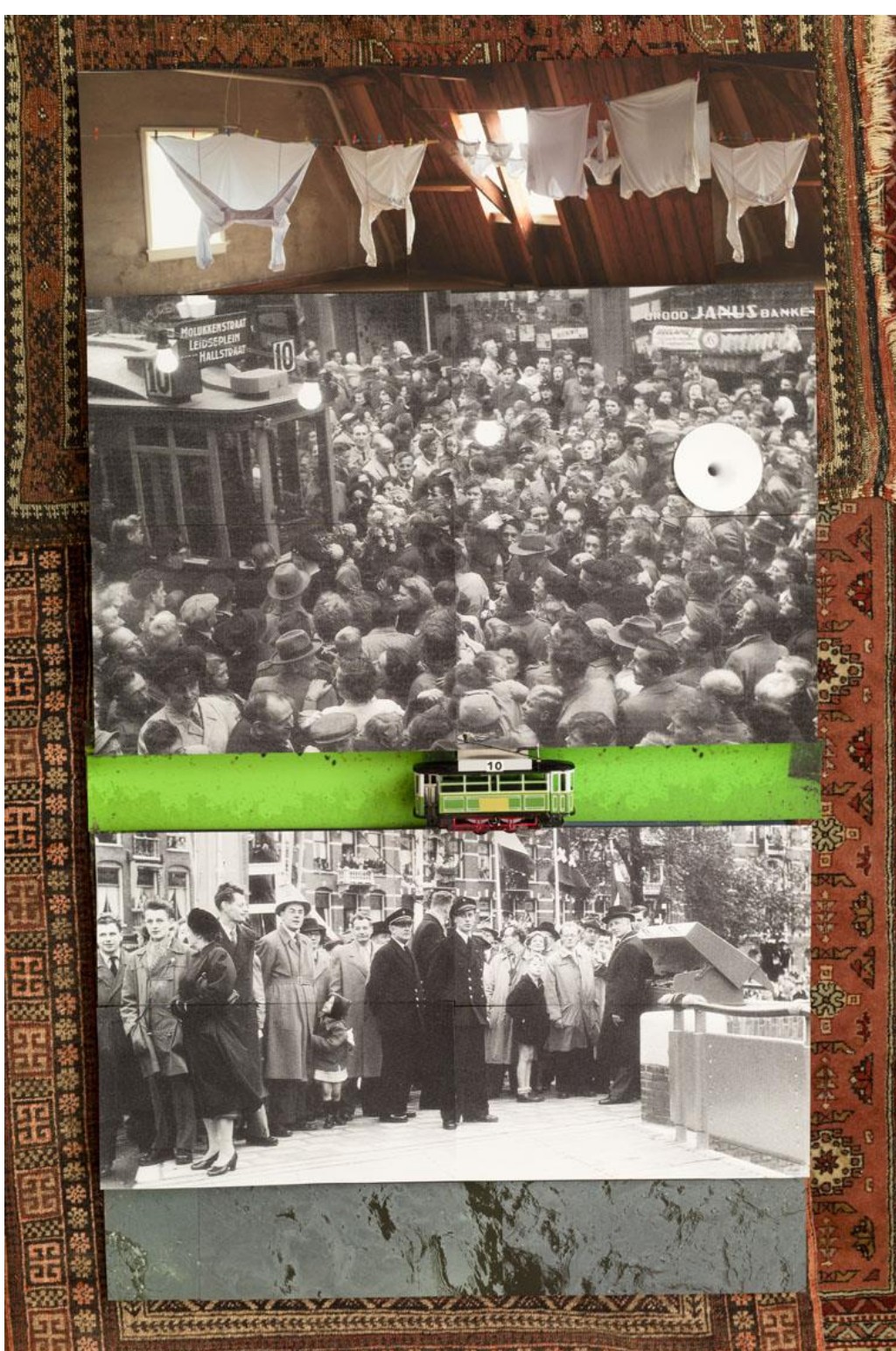

**Figure 4.** The arrival of the tramway in the neighbourhood in 1952.

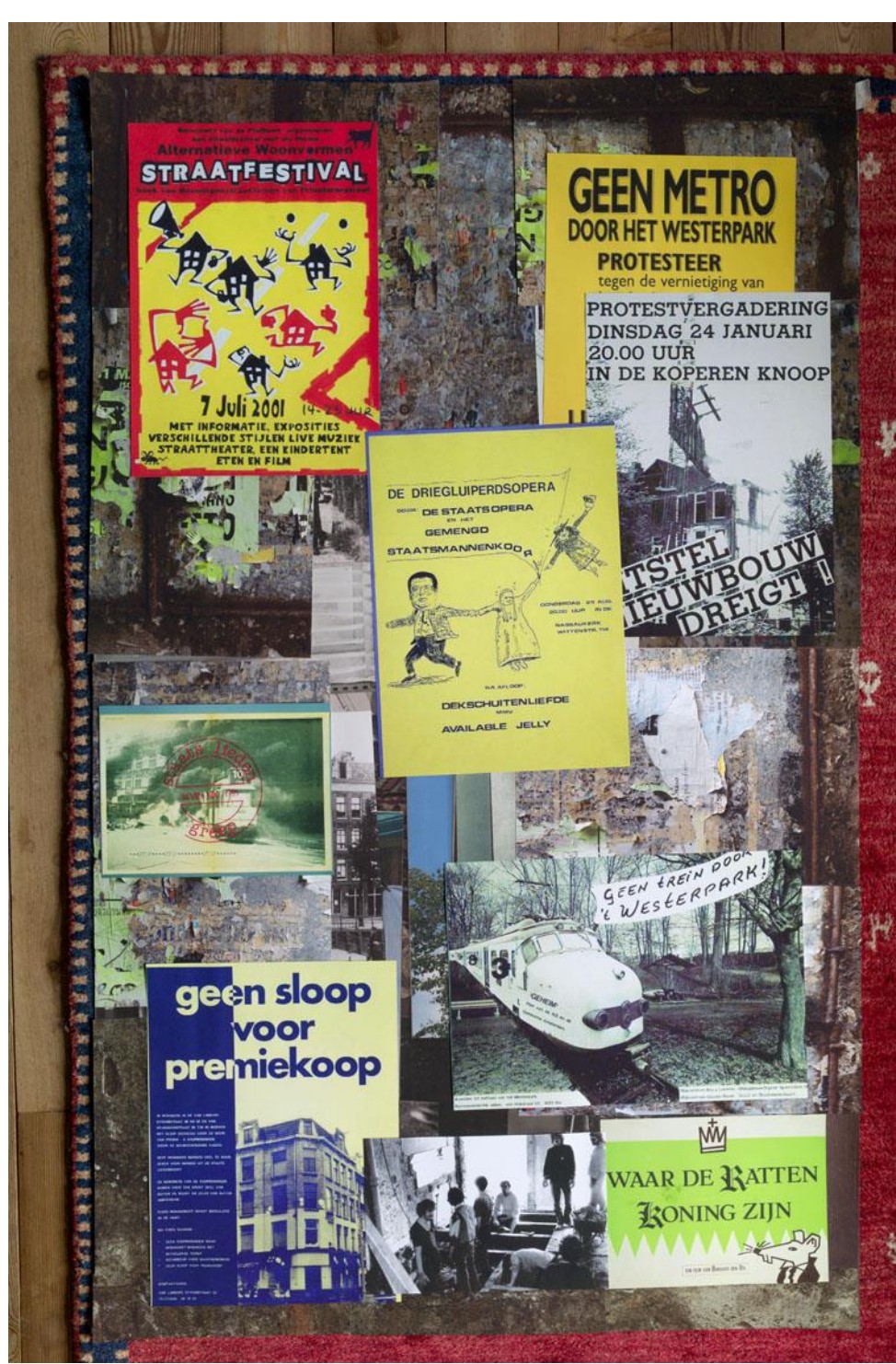

**Figure 5.** First half of the collage on the period of the squatters (1970s–1980s). They caused trouble but also preserved houses and built a social structure similar to a place where you can borrow any tool you need to fix something in your house for almost nothing, which still partially exists.

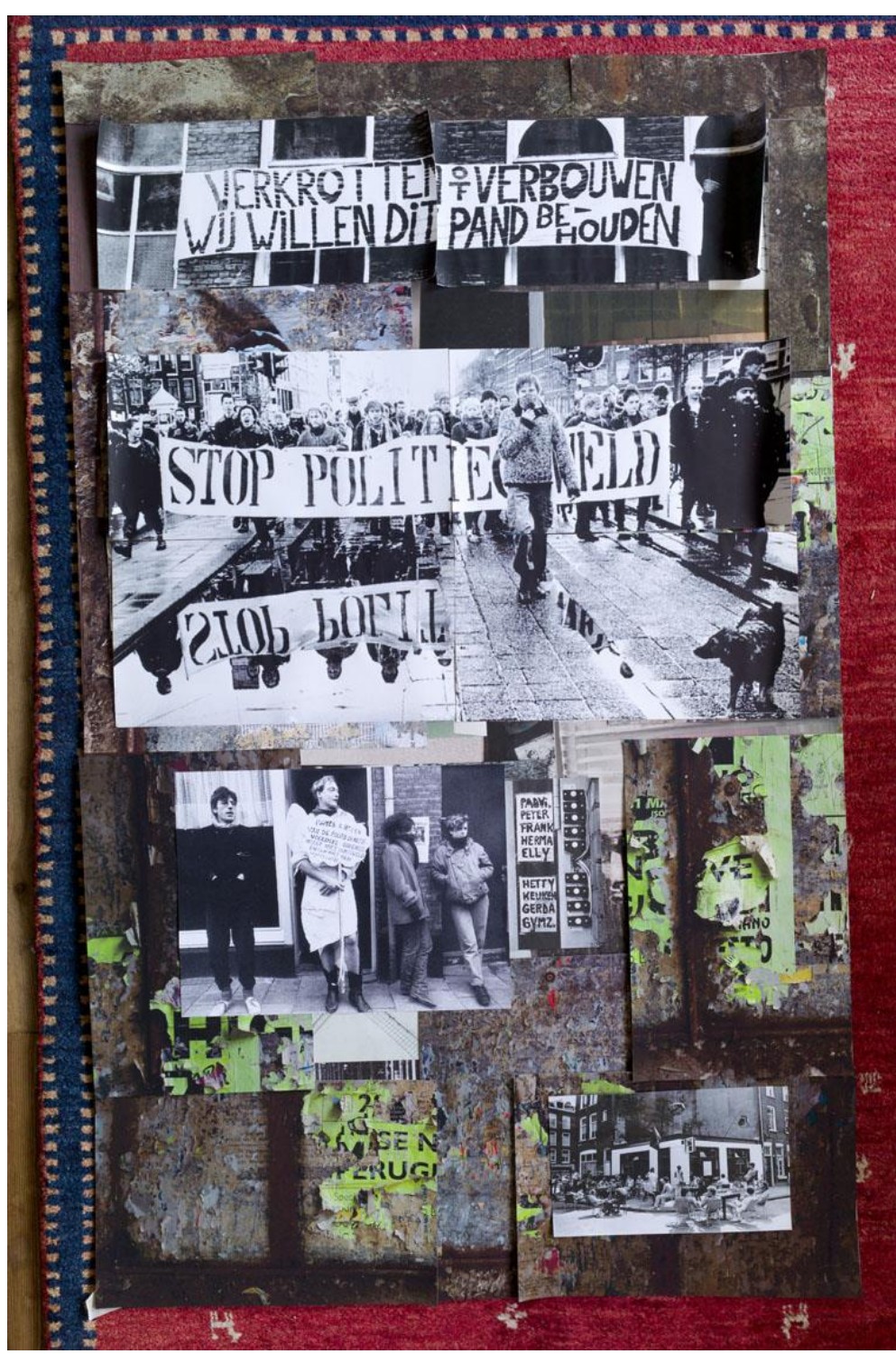

**Figure 6.** Second half of the collage on the period of the squatters (1970s–1980s). The angel was a worker from the church and is holding a sign saying "Angels do not like violence or speculation".

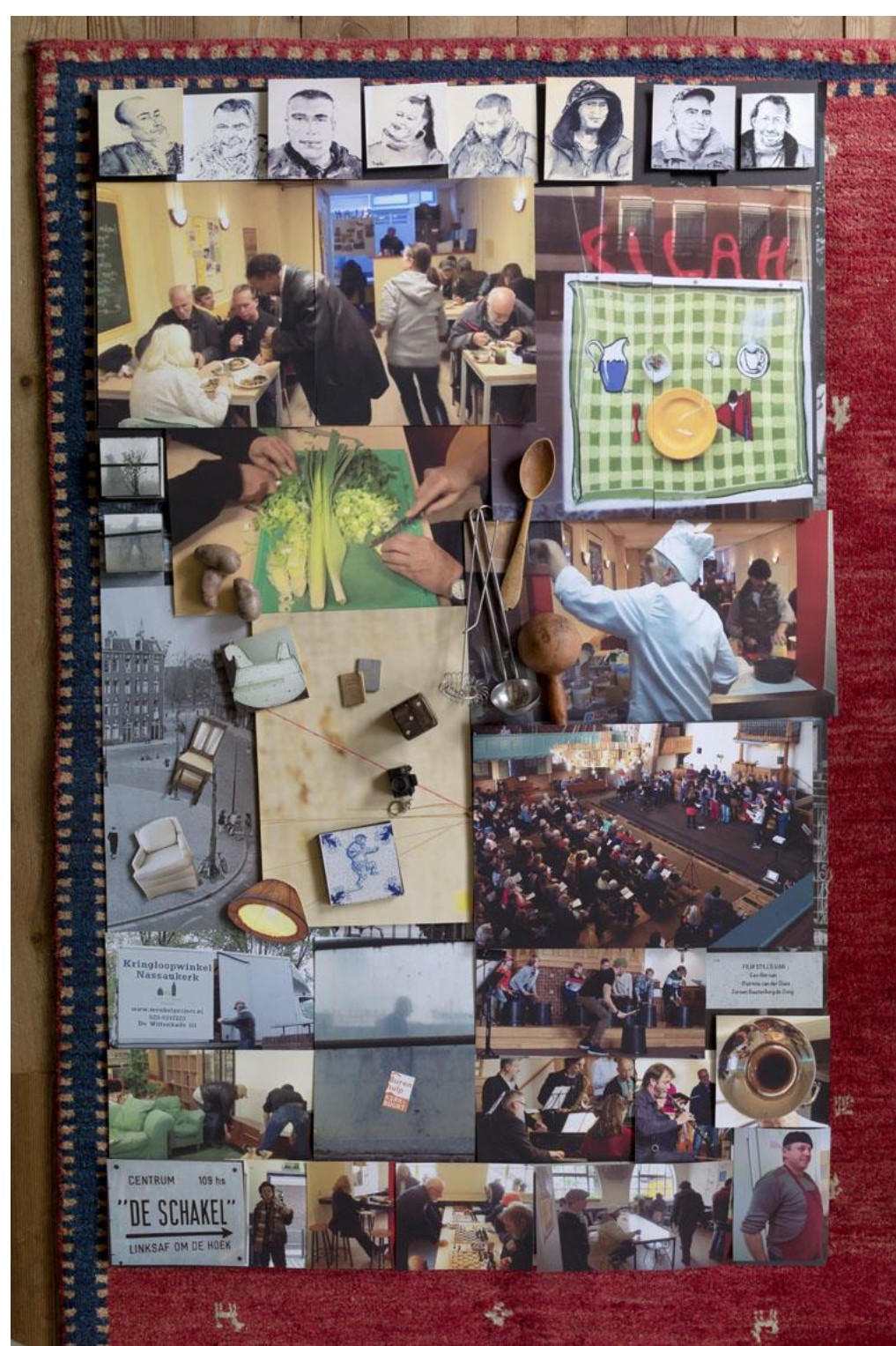

**Figure 7.** First half of the collage concerning the activities of the church in the neighbourhood from the 1980s to the present day: restaurant Filah, walk-in centre De Schakel, second-hand shop for furniture, and church services.

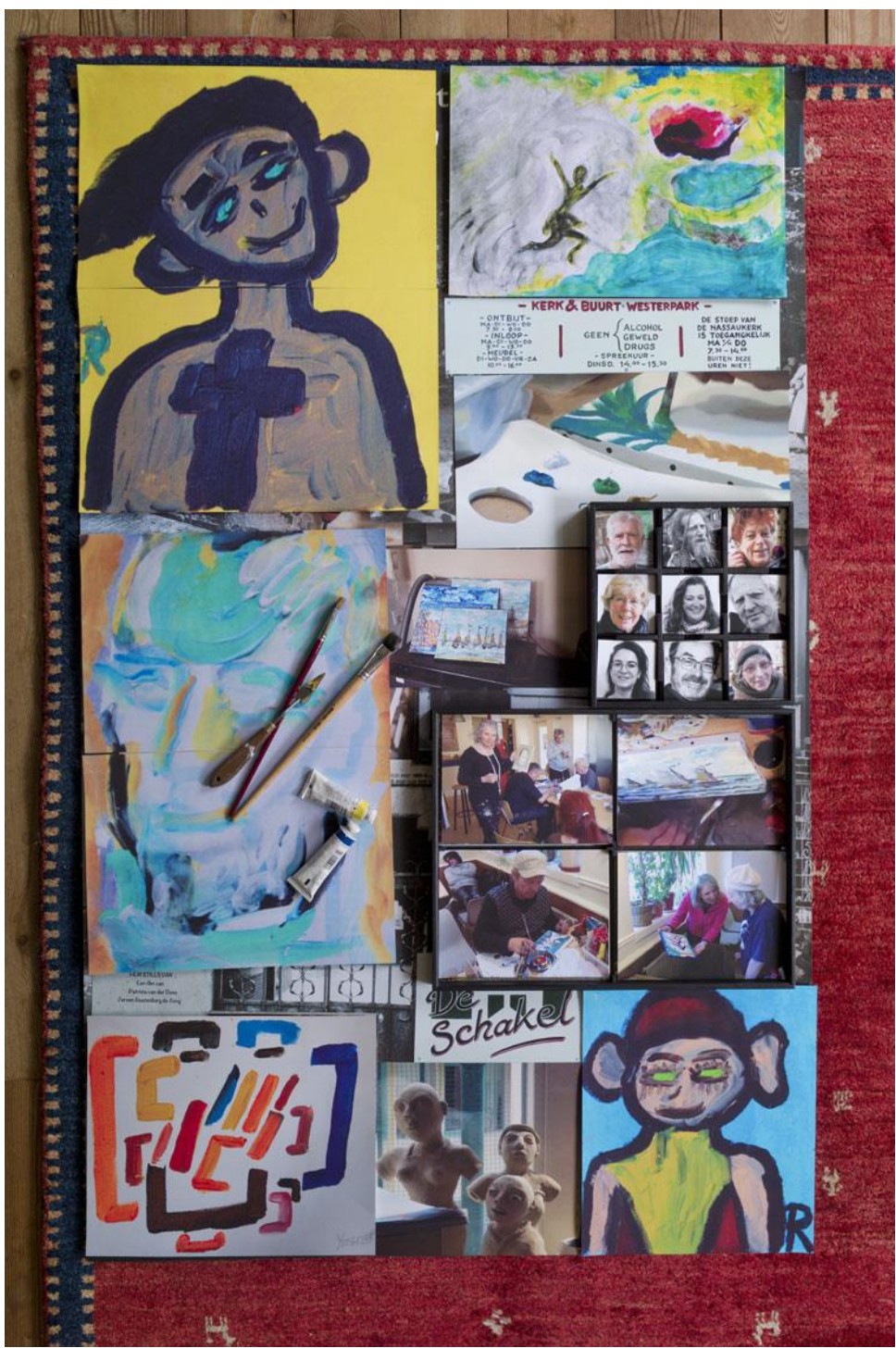

**Figure 8.** Second half of the collage concerning the church's activities in the neighbourhood from the 1980s to the present day. The painting workshop with homeless people was part of the exhibition.

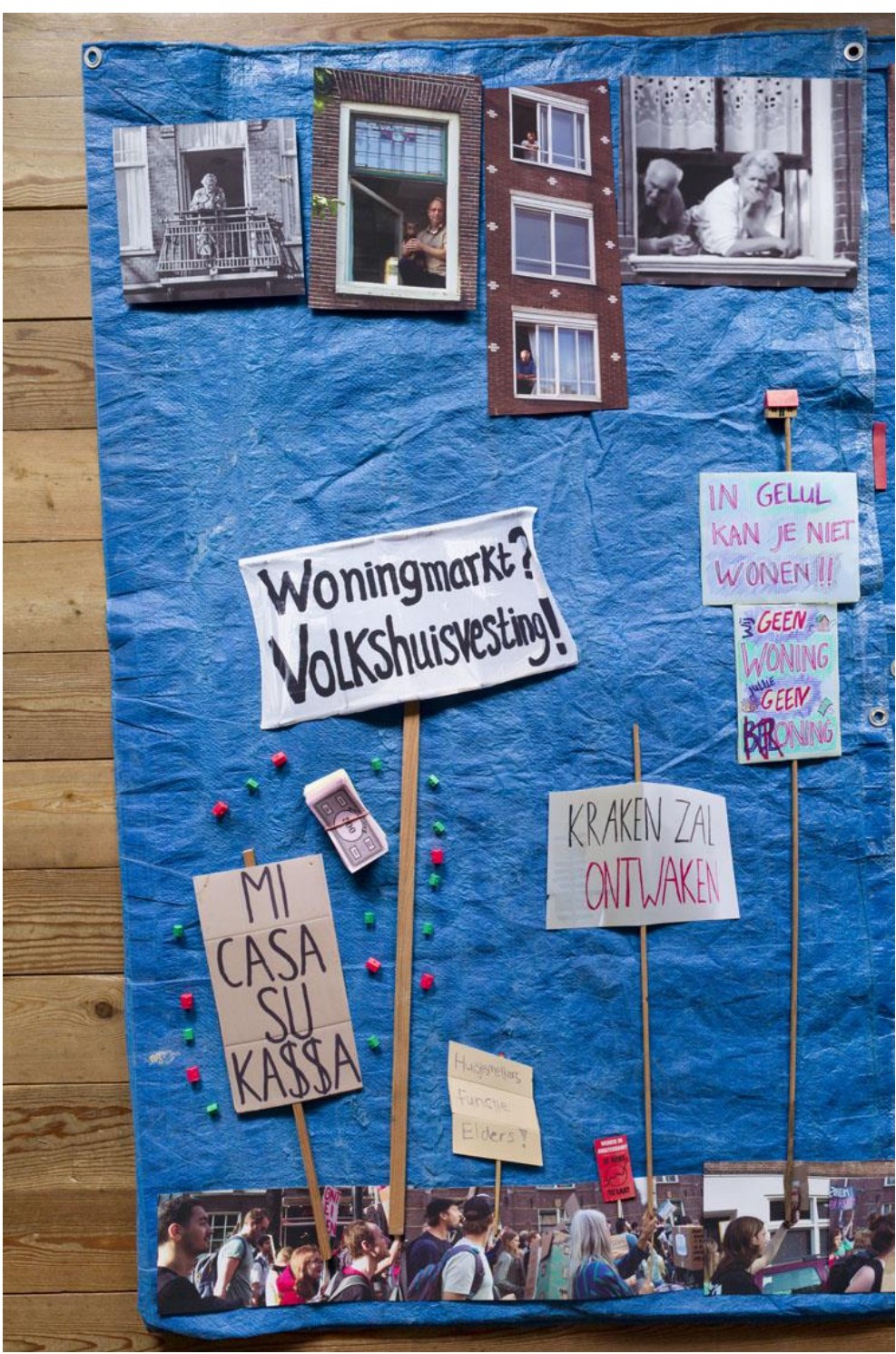

**Figure 9.** First half of the collage focusing on problems with the housing market today in Amsterdam, inspired by a big demonstration in September 2021. Some of the initiatiors of *A(t) home in the Staats* are participating.

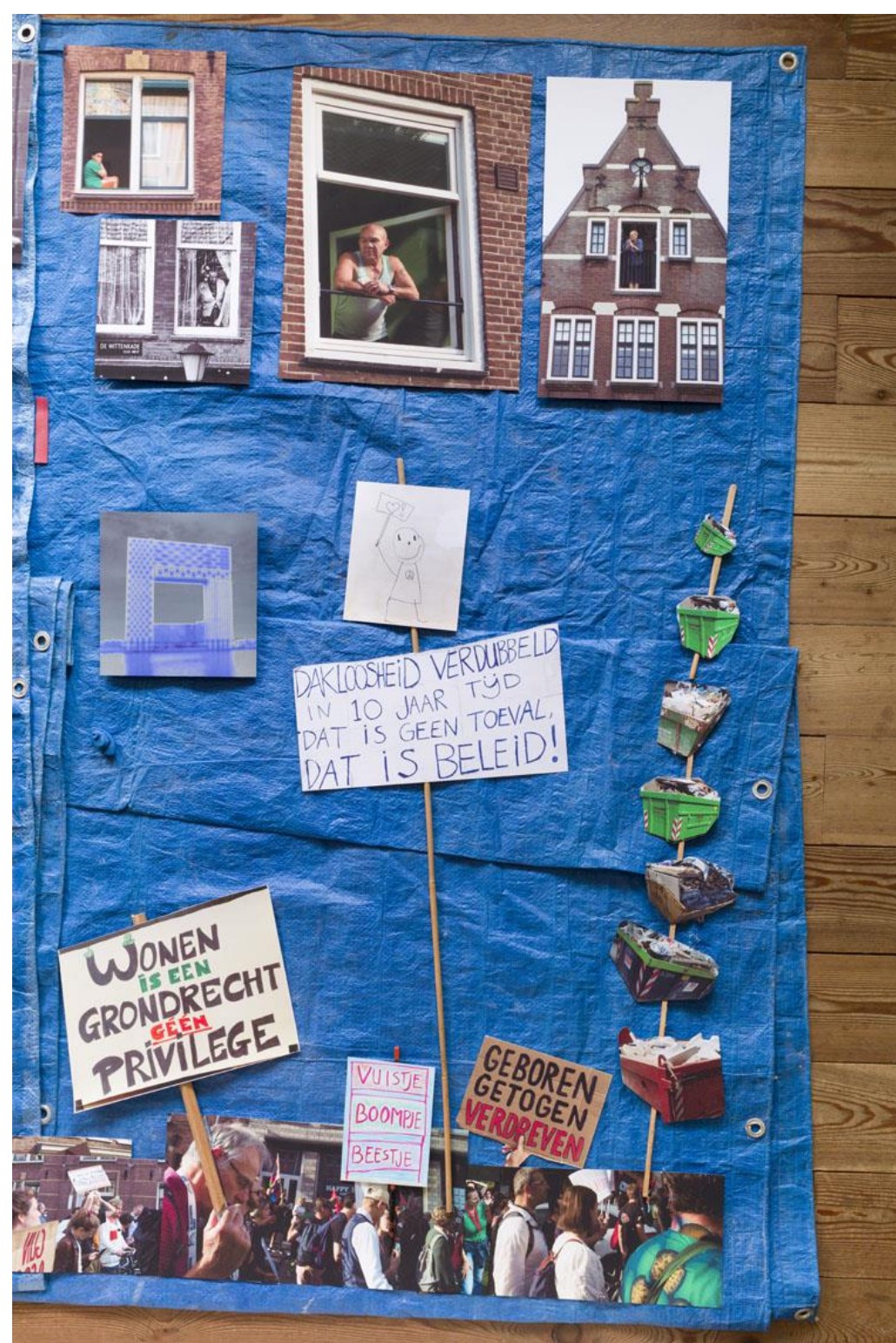

**Figure 10.** Second half of the collage regarding the problems with the housing market today in Amsterdam, inspired by a big demonstration in September 2021.

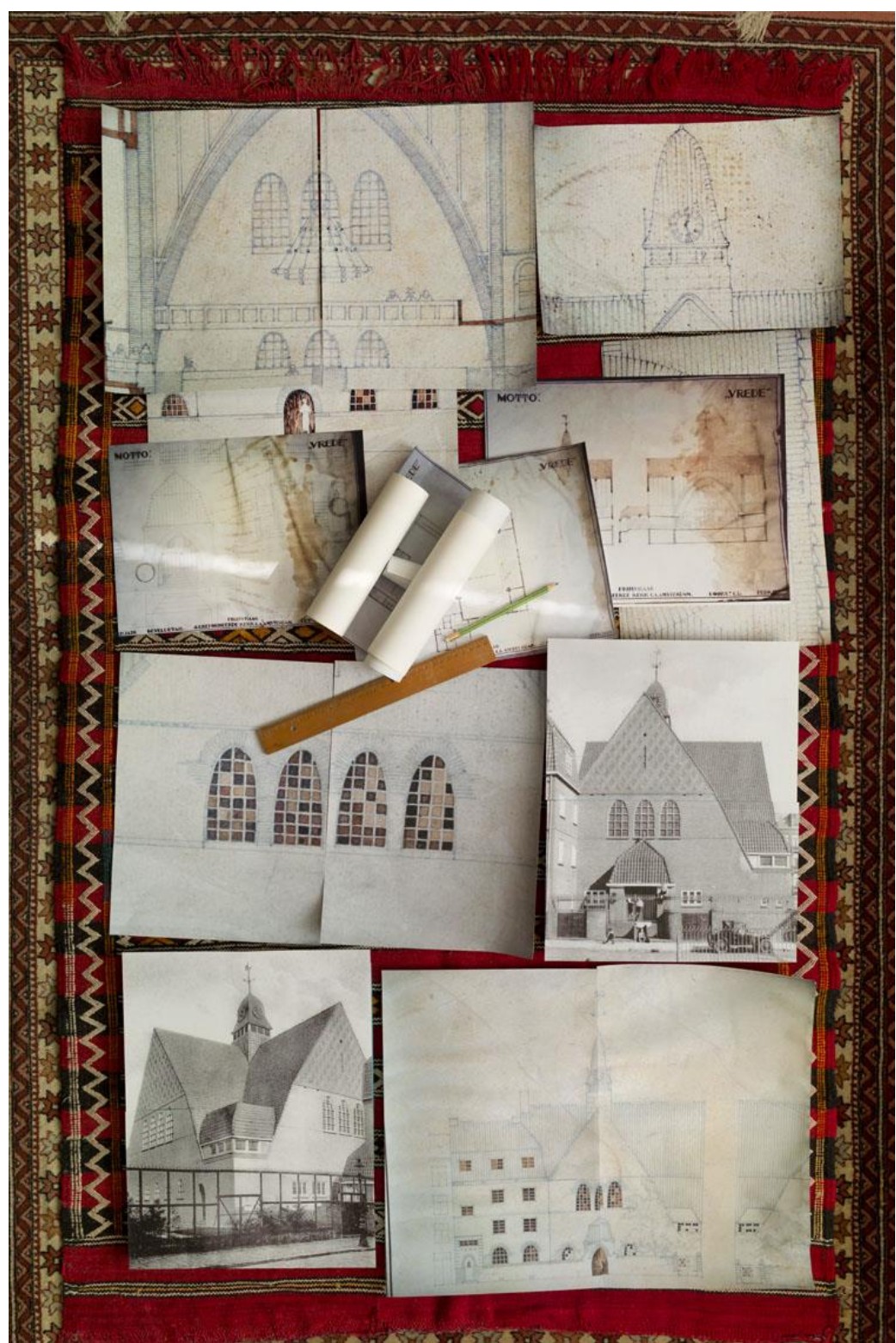

**Figure 11.** Collage on the design competition for the building of the church during the 1920s, won by architect H.G. Krijgsman. All collages: Philippe Velez McIntyre, 2021. More information: https//www.philippemcintyre.com accessed on 14 June 2023.

During the exhibit itself, the collages and portraits were displayed on huge posters in the church, inviting visitors to react and share their own memories (Figure 12). For the same purpose, intimate spaces were created and decorated with traditional, old-fashioned

wallpaper and furniture fitting the lifestyle of the "pre-squatting" and "pre-gentrification" labour population. In these tiny "living rooms", visitors were invited to have tea and chat. While the exhibit was open (four days a week for a couple of hours), old and new activities were brought together in the church: a neighbour who had joined the preparation group offered creative writing workshops. The painting workshops with homeless people from the church and neighbourhood were now in the church and open to everybody. The paintings were added to the exhibit and made available for sale. The written stories were recited on special evenings called "neighbours retelling". A "neighbourhood cantata" was performed twice, once in the church and once in a school. It had been written by neighbours on the theme "at home in the Staats" and composed by one of the musicians of the congregation. A percussion workshop with (break) dancing was organised for children from schools in the neighbourhood. Information about, and reports on all activities were brought together through a specially designed website.[32] A participant in church and Neighbourhood spontaneously started to make short films during several activities, such as the painting workshops[33]; interviewing passers-by on the streets,[34] and homeless people[35], telling how they felt at home and what they liked or disliked in the neighbourhood. Finally, one of the neighbours led guided tours through the neighbourhood.

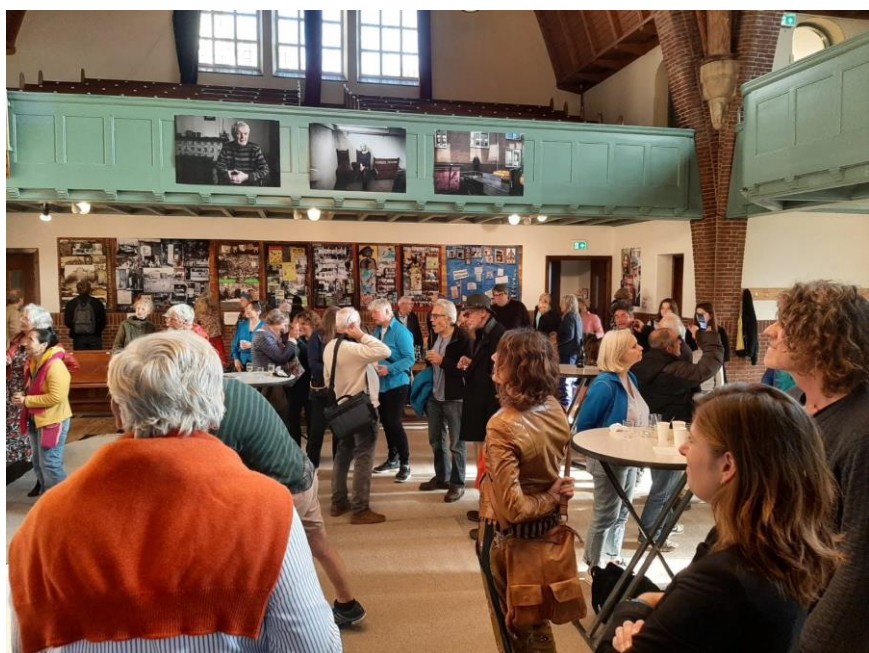

**Figure 12.** Opening of the exhibit on 9 October 2021. Picture: E. Meijers.

The musical, literary, photographic, and visual methods all aimed at collecting stories and stimulating people to exchange and, as the distributed leaflet said, to "look at the neighbourhood through the eyes of others". The political and activist aspects of the exhibit received attention during the vernissage of the exhibit, when local politicians and (active) neighbours were invited to speed date. At the closing debate, with representatives from neighbourhood organisations, local politicians, and the church, findings were proposed, questions raised, and points of action discussed.

Concluding this section, we can say that there was a correspondence between artistic and diaconal ways of working. This correspondence was defined by the four features above, which fostered openness in many ways and left space for others to connect to the ongoing processes. In addition, the use of artistic methods worked as an invitation to participate, and they attracted people from different backgrounds to participate in the exhibit.

### 3. Belonging to the Neighbourhood: Outcomes of the Exhibit

What did we learn about local residents' sense of belonging and the issues they were wrestling with? First, the flood of stories, many of which we could not capture as they occurred between visitors to the exhibit and during informal meetings, had a value of their own: they connected different people for a moment, bringing memories and desires to life. The same is true of the painting, singing, dancing, drumming, walking, and writing, as well as other evasive experiences such as little smiles, eyes lighting up, memory flashes (somebody screaming out: I have such bad memories in this church!), and moments of surprise that cannot be documented. Yet they all contributed to the recognition of the experiences of "those who do not generate money, are useless, and are overlooked", as Saskia Sassen has put it. (Meijers 2015, p. 23). I will illustrate this with an example. Mariette (this is not her real name) is an intelligent woman. She was born outside the Netherlands and lived in various European cities before coming to the neighbourhood in the eighties. She became part of the social movement of the squatters and raised her children in the neighbourhood, which she considers to be her village. Yet, life was hard on her, and now she lives in poverty and suffers from debt and alcohol problems. She was already known by Church & Neighbourhood because she sometimes had breakfast or coffee at the drop-in centre. During *A(t) home in the Staats* she joined the writing workshop. At the second storytelling evening, which focused on stories of the street[36], she presented her story in the church. This in itself was a big step—being willing to talk in front of an audience as somebody who visibly belongs to the "shabby" side of the neighbourhood. Her story about her own life in the neighbourhood revealed a literary talent and touched the listeners. Something happened during her performance, both to herself and the audience: clothed as the poor woman she had become, with all the signs of a hard life in her appearance, a fresh and creative spirit emerged. Looking back after the exhibit, one of the initiators of the evening said, "She has experienced that she is one of us, and now she is a respected volunteer".[37] She visits the church from time to time and shares the Table of the Lord with the congregation. Her problems are not finished, but her feelings of belonging to the neighbourhood have increased and are not just past memories. She was not the only one who became a volunteer through *A(t) home in the Staats*. A year later, other participants were still active. They are not many, but the project changed them from anonymous residents to active neighbours. This could be called the most tangible result of the exhibit.

Can we answer the research question of the group? Do people feel at home in the Staats? At first sight, the answer is yes: both the spontaneous answers on the streets and the more in-depth interviews give an image of an almost village-like community, where people greet each other on the street (they feel 'seen'). One interviewee expressed this as follows: "The strange thing is when you go under the tunnel, you arrive in another world: ( . . . ) The streets are lively, and I like the atmosphere. I greet everybody in the street even if I do not know anything of half of them" (Dorssers and Meijers 2021, pp. 30–31). The residents meet each other in the park, cafés, squares, the church, and through many activities that often find their origins in the time of the squatters. This period was mentioned by many interviewees as a defining period for the neighbourhood. Although for some, this was a terrible time, others have good memories. One of the long-time inhabitants told us, "This is my village and my world. The ideals of the early days—the social revolution—characterise the neighbourhood. You still see it in how people deal with each other: it is multicultural; some parts are Turkish, others Berber; there are people from Suriname; we meet all the time; the children are in the same schools; and people have to do something to make things work out; you have to fight for it. That is the *Staatsliedenbuurt* to me" (Dorssers and Meijers 2021, p. 20). The diversity in terms of colour, age, class, and lifestyle is appreciated by many interviewees, just as the *Westerpark*, situated at the edge of the neighbourhood, is mentioned by many as a place where they enjoy both nature and cultural activities (which are sometimes annoying as well).[38]

However, when one listens below this positive surface, there seems to be a precarious balance in the areas: it is "*still* for everybody", and "we have to fight to make things work

out". Based on the ten interviews published in the magazine and the two evenings where stories were shared in the church, several issues have come up that could threaten feelings of belonging. First, the situation of the housing market and rental prices have a segregating effect between wealthy newcomers and poor or lower middle-class inhabitants. "People move out of the quarter more quickly; they do not stay very long. Their relation to the neighbourhood is much looser than before" (Dorssers and Meijers 2021, p. 6). "We do not want expensive houses here ( . . . ). They are beautiful, but they change the types of shops and restaurants. I do not want Ton's bicycle shop and Ali's Market to disappear". "I would like the neighbourhood to stay mixed, but I am afraid it will not: the new places here target another public" (Dorssers and Meijers 2021, p. 47).

The second worry mentioned by almost all interviewees was the change in shops from places where you could have a chat to anonymous services. Shops have an important social function and sometimes even greater functions. For example, one of the small grocery stores knew exactly what type of tobacco an older resident, who was slowly losing his memory, always smoked and prepared it for him. When he came back ten minutes later to buy it again, the shopkeeper showed him the tobacco in his bag.[39] Yet, every year, more shops, which defined the neighbourhood for decennia, disappear. Finally, many interviewees worried about whether the appreciated diversity of the neighbourhood could be maintained. During the closing debate, one participant commented that many groups and individuals in the neighbourhood live in their own bubbles. Is this not true of the church and Church & Neighbourhood as well? Was the diversity of visitors large enough? Why did so few young people take part? Why are there no alliances with the local football club, etcetera? (Jansen 2021) Two youngsters that were interviewed for the project (they were present during the opening of the exhibit) had grown up in the neighbourhood and had now started their own fashion brand named 'Staatslieden'. Like the older inhabitants, they foresaw that expats and *yups* would take the place of the less fortunate residents. They expressed their worries with a slogan, referring to the place where they used to change the small change they had made by playing guitar in the park: "Hands off snack bar, Hannie!" (Dorssers and Meijers 2021, p. 41).

In conclusion, the exhibit and its accompanying activities showed a neighbourhood where many felt at home, but where change has been noticed, and people worry about the future. Some inhabitants "own" the neighbourhood less than others. More than anything else, the initiators wanted to create space for connection and sharing. They concluded that the exhibit had certainly had an impact on the participants, but that the direct impact on the neighbourhood in terms of overcoming segregation had been small. They saw the exhibit as just a moment in the ongoing work of strengthening the social tissue of the neighbourhood, which needs permanent attention, political action, and long-term engagement. Yet participants were convinced that the exhibit's way of working—muddled, groping, and through trial and error—contains the power of a resilient society. [40] One of the insights gained was the importance of the church building as a house in the neighbourhood, which can contribute to the flourishing of the neighbourhood despite a shrinking congregation. In this cooperative project, the building functioned as a space for cultural, political, and social events. It had given space to initiatives and audiences that do not have access to the more formal (and expensive) cultural spaces in the Westerpark. The cooperation between the Nassau church and Church & Neighbourhood continued after the exhibit in two commissions. It had been, just like the exhibit, a result of the exposure process described above. One was called the 'Nassau table' and brought together all the users of the building to reflect on its future; the other started to work on a business case for the building. It formulated four pillars to shape the role of the church building as a house for the neighbourhood: 1. give priority to people in precarious conditions to contribute to justice and wellbeing in the neighbourhood; 2. actively contribute to reciprocal relations in the area with a keen eye for cultural, spiritual, and social power differences; 3. give special attention to the cooperation between professionals and volunteers; and 4. ensure sustainability, both ecologically and financially.[41] These pillars are in line with the principles of *A(t) home in the Neighbourhood*.

## 4. Conclusions and Challenges

I will now answer the central question of this article: how can alliances between community art and diaconia as community development contribute to overcoming segregation in urban contexts? The diaconal focus on people who suffer from changes in the city and the focus of community art connecting people despite social boundaries came together in this project. Artistic methods helped to articulate the aims and structure of the project. They helped to connect people through words, music, and the visual arts, enlarging the spaces in which people could participate and express themselves. They also attracted more participants and visitors and were themselves an expression of community. They helped to "imagine, strengthen, connect, and, in some cases, renew" (van den Bergh). They stimulated "self-confidence, creativity, personal growth, and social cohesion" among the participants by developing their own stories (Matarasso 1979). The story of the engagement of the congregation in the neighbourhood was also developed and found expression in the exercise of how to be a home/house for the neighbourhood. The recognition of the value of these stories was at the core of *(T)huis in the Staats*.

However, the most important outcome was that an alliance between community art and, in this case, diaconia provided structures and spaces in which stories could be created and shared, memories evoked, and dreams and worries expressed. The structure provided had a specific form, which might seem chaotic from the outside but was actually strong inside and was informed by long years of commitment. The structure was characterised by diversity, cooperation between professionals and volunteers without specific directions from a superior structure or person, an emphasis on the process rather than the product, and stamina. The space thus created was democratic and open, providing opportunities to learn about how to handle changing societies through cooperation and connection (Dzur). In hindsight, the organising group became aware of the fact that these outcomes are not easily available and might not have been reached if the group had not already had networks, personal relations, and theological and social commitments that had been practiced for over forty years but had nevertheless always remained fragile.[42] If we wanted to promote diversity, we had to be diverse. If we wanted to promote connections between people from different "bubbles" and classes, we had to work in a way that left open spaces for others to step in and connect with us. If we wanted those who usually stay invisible to be seen and recognised, we had to accept their experiences and ways of moving. In that respect, the exhibit can be seen as a way of maintaining and practicing new relationships and commitment.

This conclusion raises several intertwined theoretical and practical challenges for alliances between community art and diaconia in urban contexts. I would like to conclude this article by briefly mapping these challenges.

### 4.1. Engaging with the Unknown

The first challenge is the permanent engagement with uncertainties, chaos, and one's own lack of knowledge. It is a deeply diaconal challenge to engage with others—people different from ourselves—in situations in which we, or they, are often not in control. It implies a constant struggle with each other and with ourselves to make connections without claiming the other or forcing the other into a connection we have defined and conceived. Claire Luise Radford (2020, pp. 64 and 66) points out that arts-based research can challenge dominant theologies, especially with regards to the way people are represented and their experiences interpreted. The recognition of the unknown or uncertain plays a vital role in generating new theological understandings about how God is present in the city.

In Diaconia, this is related to the great commandment to "love your neighbour, who is like you". This comes with the constant need to realise that our longing for wholeness connects us, but that it can only heal us if we recognise that we cannot manage or control it.[43] It is rooted in the longing for God's love and compassion, which are so much greater than ourselves and yet are to be found in our fragility and pain. Community art as a restless art (Matarasso 2019), with its doubts, intentions, and ways of sharing and constantly

redefining experiences in an uncomfortable process that affects all participants (Brueguera 2010), can help to deepen our commitment and learning.

*4.2. Storytelling and Power Relations*

We have seen the power of storytelling (both in words and through music and visual arts) in this project for community building and how it can strengthen the agency of the storyteller because her/his/their experience is recognised and valued by others. As storytelling has such a central place in the alliance between diaconia and community art, it needs to be critically studied in order to gain a deeper understanding of the risks and possibilities of finding, creating, and sharing stories. How are stories constructed, and when do they receive attention? When are they valued? What are the languages of the city? As John Klaasen (2019) has highlighted, the central issues here are power and agency: who tells the story, and what is the position of the storyteller in regards to the community? This is especially important when the position of the storyteller in the community is changing due to urban developments.

*4.3. Ownership and Collective Decision Making*

*A(t) home in the Staats* investigated if and how inhabitants felt they belonged to the neighbourhood. The underlying question was: who owns the neighbourhood? We found that while some people feel more comfortable and move more easily through the neighbourhood than others, other people constantly struggle to make a place for themselves. Artistic methods influence the findings, and the decision-making structures influence the way people engage with a project such as this. Church communities often follow deeply engrained models of decision making and project management: the church council, the minister, and other offices all have their own place and authority. The dynamics of changing cities and fading congregations call for a critical revision of these familiar structures. The diversity of participants was an important feature of this project, and in cities in which churches are shrinking more and more, they will have to cooperate with others in order to have meaning. Dealing with a diversity of motives, perspectives, and levels of participation is a vital challenge in projects such as these.

What kinds of decision-making structures hinder or enhance cooperation, diversity, and community building? Who owns a project that is set up by Christians and people with other outlooks on life? How can we regulate common ownership? What structures offer space for the qualities of both participating volunteers and professionals? How can we stimulate cooperation and exchange between persons of different ages, genders, colours, classes, generations, and lifestyles, such as those who have a home and a social network and those who do not? These are very practical questions, which receive different answers in each context; nevertheless, the importance of these questions is often overlooked or avoided since they touch the power structures of churches and organisations. As we have seen, the alliance between diaconia and community art confronts congregations and communities with the need to change. The perpetuation of old structures, discourses, and worldviews, hierarchies, and the distribution of money might make it very hard to contribute to the reality of changing cities. Insights from others, such as those of specialists in community art, cultural participation, and social and democratic development in cities, can be relevant here. Matarasso (2019, p. 17) has pointed to the democratic values of ambiguity and crossing borders. In order to create spaces in which change can occur, both community art and diaconia depend on methods that are open, receptive, and critical towards "facts", "discourses", and dominant representations of society. One concrete question regards the way that money is distributed to local churches based on the number of official (baptised or confessing) members. One can wonder if this is helpful in a situation where the active, actual community is a lot larger than just registered formal church members. The questions De Roest (2020, p. 196) raised as important for action research do not only concern projects such as *A(t) home in the Staats*. All those who want to be involved in changing urban areas in a meaningful way have to ask themselves: What are our habits and customs, and can we

revise them? What are the control structures that prevent us from being change agents in our own professional practices?

### 4.4. Witnessing the Dynamics of Visibility and Invisibility

Saskia Sassen envisages a role in which artists make visible what and who have been made invisible by political and economic powers. Through their work, diaconal practitioners often do the same as they engage with groups that are invisible in the formal structures of care in our society (Meijers and Tromp 2022). Sharog Heshmat Manesh (2022, pp. 242–48) coined the term "witness" for cultural professionals: they dig out yet unarticulated experiences, help convert them into full stories that can be shared and recognised, and then present them to a public, thus making them visible and real. By doing this, they contribute to the restoration of injustice and pain. The French theologian Jacques Ellul defines Christians living in cities as "witnesses of justice, put in that position by the Word of God"[44] (Ellul 2003, pp. 147–48). In this case, witnessing has to do with staying faithful, however terrible the urban practices might be. As witnesses, Christians must stand in solidarity with other urban citizens. We must give all we have for the flourishing of the city, which, to Ellul, means that we must not live from a closed worldview but rather live in expectation of the coming reign of God. In the meantime, it is our task to awaken expectation, restlessness, and hope.

**Funding:** This research received no external funding.

**Informed Consent Statement:** Informed consent was obtained from all subjects involved in the study.

**Data Availability Statement:** The data presented in this study are available on request from the corresponding author. The data are not publicly available due to privacy and ethical restrictions. Restrictions apply to the availability of another part of these data, as they were obtained from several third parties (archives of local churches, private members of the church and of local organisations; they are available from the corresponding author.

**Conflicts of Interest:** The author declare no conflict of interest.

## Notes

1. Examples in Europe: Florian Malzacher and Jonas Staal (eds.). 2021; Das Zentrum für politische Schönheit: https://politicalbeauty.de/ (accessed on 20 March 2023); Welfare State International, https://www.welfare-state.org/pages/aboutwsi.htm (accessed on 20 March 2023) and the work of Bart Stuart and Klaar van der Lippe at https://www.burospelenblog.nl/ (accessed on 10 April 2023).

2. The title of a recent diaconal publication on community development by Regnum Books (Haugen et al. 2022)

3. In the Dutch context, I have found only two publications: Albert Ploeger (2002) and Nico van der Perk (2013).

4. I am grateful to Wiel Dorssers for suggesting this translation.

5. Other religions: thirteen percent Muslim and one percent as Jewish, Buddhist or Hinduist. *De Staat van de Stad Amsterdam X, 2018–2019*, OIS. https://onderzoek.amsterdam.nl/publicatie/de-staat-van-de-stad-amsterdam-x-2018-2019 (accessed on 11 April 2023).

6. Estimated guess of the organisers, based on visitors and participants in the exhibit (Focus-group 17 April 2023).

7. This focus-group took place on 17 April 2023.

8. See my article 'Walking in the neighbourhood' in the issue of *Religions* on 'Christian Congregations as Communities of Care' (forthcoming in 2023).

9. This historical information is partly based on interviews with inhabitants, and partly on literature and archive research done by participants, one of whom was a masters student in history. The outcome is published in the catalogue of the exhibit *A(t) home in the Staats. Expositie en buurtgesprekken vanuit de Nassaukerk*, 2021.

10. Interview E. Meijers and J. Buning with Paula Irik, minister of the Nassau church during those years, 13 August 2019.

11. All Dutch citations are translated by EM.

12. https://www.kerkenbuurtwesterpark.nl/ (accessed on 11 April 2023).

13. A term coined by urbanist Zef Hemel. See https://zefhemel.nl/stad-zonder-kunst/ (accessed on 19 May 2023).

14. CBS: Het aandeel sociale huurwoningen blijft dalen. *Het Parool*, 20 October 2022.

15. Statistics of Amsterdam area's: https://allecijfers.nl/wijk/staatsliedenbuurt-amsterdam/ (accessed on 14 April 2023).

16    'Stadsdeel wil overlast Westerpark beteugelen', *Het Parool* 14 September 2018 and *Staatsliedenkrant*, April 2023.

17    Source of this story: focus-group with the organisers, 17 April 2023.

18    Policy paper: *Inspirerende presentie–een nieuwe rol voor de kerk in de stad. Beleidsplan Nassaukerkgemeente* 2018–2022.

19    Policy paper 2018–2022.

20    Policy document Nassauchurch 2017–2022.

21    Fokje Wierdsma, *Report on exposure in the Nassauchurch 2016–2019*, February 2019.

22    For example 'Blik op de buurt. Ons werk in relatie tot de buurt', *Over De Brug*, January 2019, in which members of the congregation were invited to participate in 'exposure walks' in the neighborhood.

23    This center "supports and trains professionals and volunteers in experienced based learning, who support people living in circumstances in which their humanity is under pressure". https://korschippers.nl/overons/ (accessed on 13 April 2023).

24    Minutes and reports of the exposure group between 2017 and 2020.

25    Three artists (Philipe Velez McIntyre, Bart Stuart, Klaar van der Lippe), the minister Klaas Holwerda (who was only present the first time. He left the group in the beginning of 2020 and the congregation in the summer of 2021), three members of the exposure group of the Nassau church (Gerrie Willemse, Jaap Buning, and myself) and two members of the organisation 'Church and Neighborhood Westerpark' (Jette Uittenhout, Dick Jansen). (source: minutes 16 September 2019 and 2 February 2020). Photographer Peter Valckx, Writer Wiel Dorssers, social worker Hilde Dijkstra, designers Mark Schalken and Paulien Jansen en student history Rachel Meijers joined later.

26    Minutes (A)t home in the Staats, 16 September 2019.

27    Minutes of *(T)huis in de Staats*, June 2021.

28    Jaap Buning, final report *(T)huis in de Staats* for one of the funding bodies, April 2022.

29    Evaluation document by the group of initiators, April 2022.

30    Focus-group 17 April 2023.

31    Minutes interview-group 30 November 2020.

32    https://thuisindestaats.nl/ (accessed on 14 June 2023).

33    Jacqy Wolters made the interviews for 'Knooppunt tv': https://www.youtube.com/watch?v=JquxoJXH1xM (accessed on 14 June 2023).

34    https://www.youtube.com/watch?v=qiuc8IVgv_o&t=79s (accessed on 14 June 2023).

35    https://www.youtube.com/watch?v=1TZzrka9Hz0&t=7s (accessed on 14 June 2023).

36    https://thuisindestaats.nl/programma/buren-vertellen/ (accessed on 19 April 2023).

37    See note 30 above.

38    These conclusions are based on the following sources: Jaap Buning, final report *Thuis in de Staats* for one of the funding bodies, April 2022; Dick Jansen (2021); Dorssers and Meijers (2021).

39    Story shared by Dick Jansen during the closing debate on 26 November 2021.

40    See note 30 above.

41    Vision Document of the commission 'Management building Nassau church', January 2023.

42    See note 30 above.

43    This aspect is developed more deeply in my article 'Walking in the neighbourhood' in the issue of *Religions* on 'Christian Congregations as Communities of Care' (forthcoming in 2023).

44    Translated from French by E.M.

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
