# Peer review of "Belonging to the City: Alliances between Community Art and Diaconia as a Means to Overcome Segregation in a Gentrifying Neighbourhood in Amsterdam"

_religions, doi:10.3390/rel14060811_

Round 1

Reviewer 1 Report

This research is an attempt at a new connection between community art and diakonia in Amsterdam. Multiple gentrifications of neighborhoods and cities trigger crises of social community. This study shows how diakonie efforts to address these crises can creatively meet with art. The study successfully demonstrates the possibilities of a new combination of local, artistic and diakonie. 

Author Response

Thanks for your review. You have indicated several things can be improved, but I didn't find any details about the improvements. I will go through the text again, now that I have a bit more distance thanks to the waiting time, and try to improve on the issues mentioned.

Reviewer 2 Report

I enjoyed reading this article because it provides so much food for thought about the role of the church in urban areas. The article describes an experimental community-based Christian art exhibit at a declining church in a newly gentrified neighborhood of Amsterdam. The conclusions reached are heartening, particularly for any interested in thinking through how art can be a useful tool to serve those in need.

While I enjoyed the content of the article thoroughly, at times I found the language difficult to understand and would suggest a thorough copy-edit by a native English speaker.  Below are some areas in need of editing:

Abstract, line 1 and throughout: capitalize Protestant

Abstract, line 12: "The" starting point of these methods...find another way of saying "starting point for reflection" in the sentence...the sentence could use some refining.

Line 26: "Art expresses the realities in which we live..." and the rest of the sentence is unclear/awkward. This seems to be a translated quotation from a Dutch source, so paraphrase it with greater clarity.

In this first paragraph, could the office of Deacon be clarified in another sentence? Could you provide a dictionary definition of the theological term or some historical context of how this role was defined in the past? I ask because I'm not sure all readers will be so familiar with "Diakonia". How did deacons traditionally seek to help the poor? What were the original aims of the office - feeding the poor, healing them? This will help the reader understand the new direction of the office in the art project.

Line 36 - this first sentence is awkward/wordy. The definition of community art beginning with line 40 is rather general. I know it is a quotation of a famous artist, so maybe add a sentence that refines the ideas in your own words? You end the quote with "It affects us" and I would ask...to what end? To what end in your project? Later on (line 84) you admit that "community art" is not easily defined...why not? Because it is such a new development, particularly in churches?

Line 97 - "Alliances...part of the Christian tradition forever." This sentence is too general. Forever? Maybe Exodus 31 with the tabernacle?

The next sentence about "art has been used to express, teach"... This blanket statement needs some art historical context, particularly if the argument is about a new role for art in diaconal practices.

line 122 "only fifteen percent"...of the population

Line 134 - The value didn't lay...this is awkward.

Line 144 - "More...what was going on." This sentence is awkward. The next one needs grammatical refinement.

Line 150 - "I will look at both functions, which cannot be separated." Why not? The next paragraph is confusing in terms of tenses and heavy-handed writing with general statements that are repeated again and again without justification. The use of I and we and participants - this needs greater clarity.

Line 204 - areas

Line 220 - Is the Church as female person or a thing? Is this a traditional reference to Ecclesia as a woman? I'm confused.

Line 346 - bowl of vegetable soup

Line 349 - vary use of "neighborhood" in the sentence

Line 392 - decisions are made, not taken (this is true for line 679, too)

Line 406 - temporary??

Line 535 - this paragraph of quotations without introducing them is a little confusing.  line 538 - "ism"?

Line 556 - the quote seems off "The changes make that people pass by more quickly?" What does this mean?

Line 623 - Is the most important conclusion really that stories were shared and worries expressed? Or could it be more like...we helped the poor by...what...allowing a shared space for healing? Develop this 

Line 648 - art-based research can enable research? what?

I'd make more of the conclusions in the paragraph beginning on line 652 because this is the true importance of your project.

Again, I think this is a terrific article! One these areas are tightened, it will be even stronger.

While I enjoyed the content of the article thoroughly, at times I found the language difficult to understand and would suggest a thorough copy-edit by a native English speaker. 

Author Response

Thank you very much for your encouraging comments! And thanks even more for the detailed comments, which will help me to improve the text. Many of them are language issues; the article will be corrected by a native English speaker, I submitted it a bit hastily. Some of your comments have to do with the special context of the Netherlands; I will clarify them. Finally, you give some input on paragraphs that are not clear or need more context or need to be developed more. These comments are of particular importance to me. I will profit from all of them while writing the final version. Thank you!